# Optimal Dietary Protein/Energy Ratio and Phosphorus Level on Water Quality and Output for a Hybrid Grouper (*Epinephelus lanceolatus* ♂ × *Epinephelus fuscoguttatus* ♀) Recirculating Aquaculture System



Xiangyu Fan [1,2,†], Hong Yu [1,†], Hongwu Cui [1], Zhiyong Xue [3], Ying Bai [1], Keming Qu [1], Haiyan Hu [2,*] and Zhengguo Cui [1,*]

1 Laboratory for Marine Fisheries Science and Food Production Processes, Key Laboratory of Sustainable Development of Marine Fisheries, Ministry of Agriculture and Rural Affairs, Yellow Sea Fisheries Research Institute, Chinese Academy of Fishery Sciences, Qingdao 266071, China; FanXYstudy@163.com (X.F.); yuhong@ysfri.ac.cn (H.Y.)
2 Marine Science and Technology, Zhejiang Ocean University, Zhoushan 316022, China
3 Huanghai Aquaculture Co., Ltd., Haibin West Road, Yantai 265100, China
* Correspondence: queencrab@163.com (H.H.); cuizg@ysfri.ac.cn (Z.C.)
† These authors contributed equally to this work.

**Abstract:** In a recirculating aquaculture system (RAS), feed is critical to the growth of fish and is the main source of nutrient pollutants in aquaculture water. An eight-week feeding trial was conducted to investigate the role of feed on the growth efficiency of hybrid grouper (*Epinephelus lanceolatus* ♂ × *Epinephelus fuscoguttatus* ♀) and water quality in a RAS. Five commercial feeds with different respective dietary protein/energy (P/E) ratios and available phosphorus levels were selected (LNLP, 31.97 g/MJ, 0.96%; LNMP, 32.11 g/MJ, 1.54%; MNLP, 36.26 g/MJ, 0.98%; MNMP, 36.53 g/MJ, 1.58%; and HNP, 41.54 g/MJ, 1.97%). The results showed that HNP had the highest growth efficiency and MNLP provided the best economic benefit. The trend in water quality within 6 h after feeding was similar among the five groups. The relative concentrations of ammonia nitrogen, total nitrogen, active phosphate, and total phosphorus reached a maximum 2 h after feeding, and the relative concentration of nitrite reached a maximum 1 h after feeding. The high P/E ratio feed increased the concentrations of total ammonia nitrogen and nitrite nitrogen. The total ammonia nitrogen concentration in HNP was much higher than those in the other treatments. The dietary P/E ratio had no significant effect on total nitrogen concentration. High dietary phosphorus levels increased the total phosphorus concentration in the water, but no significant effect on the active phosphate concentration was observed. Considering the growth efficiency, economic benefit, and water quality, it can be concluded that MNLP is the most suitable feed for RAS breeding hybrid grouper. The results of this study supplement the gap on the effects of feed on RAS water quality and provide data support for the sustainable development of RAS industry.

**Keywords:** RAS; feed; P/E ratio; available phosphorus; growth efficiency; nutrient pollutants

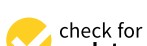



## 1. Introduction

Aquaculture—the cultivation of fish, shellfish, and aquatic animals—is the fastest-growing food sector in the world [1]. Due to the stagnation of the fishing industry, aquaculture is regarded as the only available solution to further improve the production of seafood worldwide. Traditional aquaculture systems have adverse impacts on the environment, including water quality and secondary pollution, thus causing serious damage to aquatic resources, which is not conducive to the sustainable development of aquaculture [2]. The expansion of the aquaculture industry must rely on a sustainable production model.

The recirculating aquaculture system (RAS) is system in which aquaculture water is treated and (partially) reused [3]. Aquaculture water goes through a series of biological, physical, and chemical purification processes to remove some of the pollutants in the water and then, returns to the pond or tank. RAS allows fish to be raised in a land-based, indoor, controlled environment to minimize direct interaction between the production process and the environment. Since Saeki (1958) proposed the basic theory of an RAS [4], the design of recirculating aquaculture systems increasingly improved [5–8]. RAS, once mainly concentrated in Europe and other developed regions, is now increasingly common worldwide [9–11]. In recent years, China vigorously promoted the development of the RAS industry, which is regarded as the transformative direction of green development in the aquaculture industry [12].

The RAS is becoming increasingly advanced, but its market share in aquaculture is still small. One main factor hindering its large-scale adoption is the high investment cost of RAS equipment and the long economic return cycle [13]. Simple systems are unstable and difficult to manage, whereas complex systems increase the investment costs and are difficult to put into actual production [11,14]. Feed is the main source of nutrient pollutants in RAS. Developing a suitable feeding strategy is a more direct and simple regulatory approach that can reduce the discharge of nutrient pollutants in water for an existing system, thus reducing the demand for water exchange, improving the stocking density, and reducing aquaculture wastewater discharge without increasing the initial investment cost. In addition, the nutrient composition of feed directly affects the growth efficiency of fish, which, in turn, affects the economic benefit of the RAS industry [15,16]. Therefore, the RAS industry requires more precise feeding strategies than those of traditional aquaculture  models.

Ammonia and nitrite are the main pollutants in the water of the RAS. Ammonia is converted to nitrate by nitrification in the bioreactor, and nitrite is the byproduct of nitrification. The tolerance to nitrogen pollutants varies according to fish species. The general maximum tolerance for $NH_3$ is 0.01–0.5 mg/L and that for $NO_2$-N is approximately 0.2–5.0 mg/L [17,18]. Phosphorus is also a pollutant in water, which cannot be ignored. Although fish have a high tolerance to phosphorus, phosphorus pollution of the environment threatens the sustainability of RAS [19,20]. The labile phosphate is typically considered a limiting nutrient in the water [21,22]. In the past, there were many studies on the relationship between feed formula and the concentration of nutrient pollutants in aquaculture water under traditional aquaculture. The dietary protein/energy (P/E) ratio and phosphorus level are the key factors affecting the excretion of dissolved nutrient pollutants by fish [23]. For example, reducing the P/E ratio can reduce the discharge of dissolved nitrogen waste [24,25]. When the absorption level of phosphorus was maintained below the maximum growth requirement of fish, only a small amount of dissolved phosphorus was excreted [26]. Typically, feed that contains high nitrogen and phosphorus levels leads to higher growth efficiency. Therefore, many studies formulated feed strategies for different fish species such as genetically modified tilapia, juvenile tuna, and juvenile rainbow trout that consider both growth efficiencies and nitrogen and phosphorus pollutant excretion [27–29]. RAS is enclosed and has a capacity to purify water [9]. The total amount of water is limited and so, the concentration of various pollutants in the water varies greatly after feeding. Exploring changes in water quality over time after feeding is helpful for water quality management in RAS.

Hybrid grouper, male *Epinephelus. fuscoguttatus* (Forsskål, 1775) × female *Epinephelus. lanceolatus* (Bloch, 1790), is a type of saltwater fish [30]. It can be cultured at 19–34 °C [31], and was widely cultivated in China in recent years [32]. With the advantages of rapid growth, high nutritional value, and economic value, hybrid grouper is very suitable for growing in an RAS [33]. However, there are few studies on the feed strategies for hybrid grouper, and there is a lack of feed strategies suitable for breeding hybrid grouper in RAS. The RAS feed strategy that considers both nutrient requirements and water quality improvement may be different from the traditional aquaculture model. There is still a gap in knowledge related to the effect of the dietary protein/energy ratio and phosphorus level

on the growth efficiency and dissolved nutrient pollutant excretion of hybrid grouper in RAS. The purpose of this study was to investigate the relationship between dietary P/E ratio and phosphorus and nutrient pollutants in the water of hybrid grouper RAS. Five commercial feeds for grouper with different dietary protein/energy ratios and phosphorus levels were selected to explore the concentration changes of nutrient pollutants in RAS water after feeding and the effects of different diets on RAS water quality. Under the premise of ensuring the nutritional requirements of hybrid grouper and the economic benefits of RAS, the five diets were evaluated from the water quality performance.

## 2. Materials and Methods

### 2.1. Recirculating Aquaculture System

The RAS consisted of a microfilter (Huixin, China), four-stage biofilm reactors, pipe aeration equipment, five heating pipes, six flow meters, and a UV area (Figure 1). The biofilm reactor consisted of three moving bed biofilm reactors (MBBR) using polypropylene filters as microbial carriers and one fixed bed biofilm reactor (FBBR) using plastic brushes as microbial carriers. The culture mode of nitrifying bacteria was natural biofilm formation. Each biofilm reactor had an area of 20 $m^2$ and a depth of 2 m. Ammonia was converted to nitrite and then, to nitrate in the biofilm reactors [9,18]. The ammonia nitrogen removal efficiency was approximately 44.6–55.2%, and the nitrite removal efficiency was approximately 11.9–15.3% in the whole experiment period. Five cement fish tanks were connected to the RAS, and each tank had an area of 50 $m^2$, a maximum depth of 2 m, and an effective depth of 1.5 m. The pipeline aerator oxygenates the biofilm reactors and the fish tanks. Each of the five fish tanks were heated by one of the five heating pipes to maintain the water temperature over 19 °C [31]. Flowmeters were installed at the water outlet of each tank and in front of the microfilter to monitor the water velocity of the RAS. The circulating flow of system water was 0.18 L/s. The daily water exchange capacity with the outside was about 10% of the total water volume in the system: make-up sea water entered the five tanks at the same rate (12 mL/s); 2 h after feeding, 5 $m^2$ of water was discharged from each tank at one time. A YSI multiparameter water quality meter (ProDSS, Ohio, USA) was used to monitor the temperature, dissolved oxygen, pH, and salinity of the fish tank. These environmental parameters were adjusted by regulating aeration, opening the heating pipe, controlling water exchange, and other measures to maintain stability. During the experiment, the temperature of each fish tank was kept between 19.2 and 21.8 °C, the dissolved oxygen was kept between 5.7 and 6.2 mg/L, the pH was kept between 7.60 and 7.97, and the salinity was kept between 29.1 and 31.4 ppt. The RAS was running stably for 3 months prior to the experiment.

### 2.2. Experimental Subject

The hybrid groupers (*E. fuscoguttatus* ♀ × *E. lanceolatus* ♂) selected for the experiment came from the same batch. The seedling fish were bred by Huanghai Aquaculture Co., Ltd. (Haiyang, Shandong, China). The hybrid groupers were of similar size (approximately 108.55 g) and randomly divided into 5 groups with 5000 fish per group.

### 2.3. Experimental Feeds

Five commercial feeds with different respective dietary P/E ratios and available phosphorus levels were selected and analyzed (LNLP, 31.97 g/MJ, 0.96%; LNMP, 32.11 g/MJ, 1.54%; MNLP, 36.26 g/MJ, 0.98%; MNMP, 36.53 g/MJ, 1.58%; and HNP, 41.54 g/MJ, 1.97%). The main components of the five feeds were fish meals (Table 1). The specific crude content of the five feeds was determined by experiment. The available phosphorus content was calculated using a slope ratio assay [34]. The diets were stored at −20 °C before use. The five experimental groups were named according to the feed used. The daily feeding amount for each group was 8 kg divided into two feedings (8:00 a.m. and 3:00 p.m.). The feed was weighed and recorded before manual feeding. If there was a feed surplus, the remaining feed was removed, counted, and then, calculated by multiplying the amount

of feed surplus by the average weight of feed particles to determine the total feed intake. Dead fish were immediately picked out and weighed to calculate feed coefficient [35].

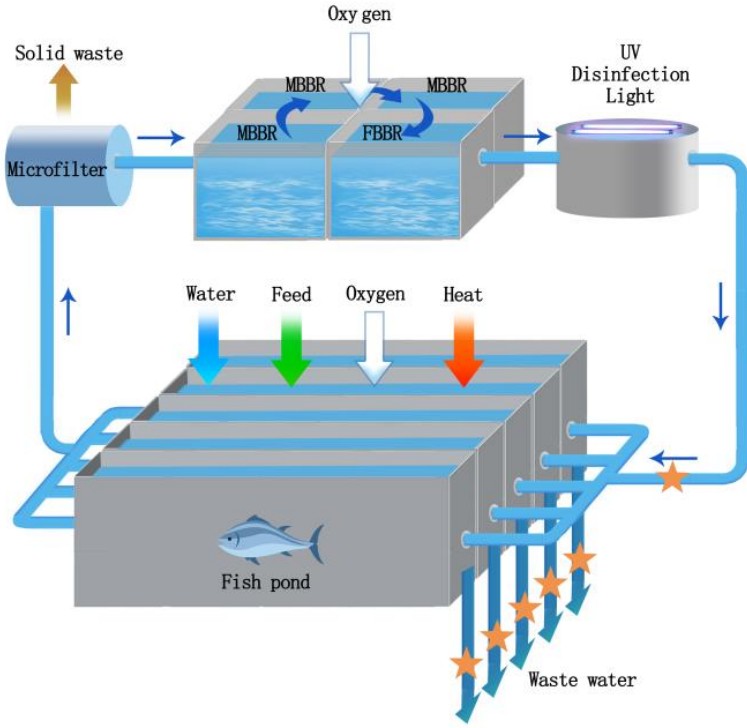

**Figure 1.** Schematic diagram of the RAS used in this study. Sampling locations are marked with stars.

**Table 1.** Composition of the experimental feeds.

| | Feed | | | | |
|---|---|---|---|---|---|
| | **LNLP** | **LNMP** | **MNLP** | **MNMP** | **HNP** |
| Main component | | | | | |
| Fish meal | √ | √ | √ | √ | √ |
| Shrimp meal | | √ | √ | | √ |
| Squid meal | | | | √ | √ |
| Soybean meal | | | √ | √ | √ |
| Fish oil | √ | √ | √ | √ | √ |
| Flour | | | √ | √ | √ |
| Minerals | √ | √ | √ | √ | √ |
| Vitamins | √ | √ | √ | √ | √ |
| Proximate Composition | | | | | |
| Crude Protein % | 49.77 | 50.31 | 53.12 | 52.99 | 58.40 |
| Crude Fat % | 12.82 | 12.71 | 9.5 | 9.49 | 9.52 |
| Total Phosphorus % | 1.48 | 1.98 | 1.51 | 2.02 | 2.43 |
| Available Phosphorus % | 0.96 | 1.54 | 0.98 | 1.58 | 1.97 |
| Crude Fiber % | 0.91 | 0.59 | 1.52 | 1.90 | 0.88 |
| Moisture % | 6.82 | 6.81 | 6.57 | 6.81 | 6.36 |
| Crude Ash % | 10.42 | 12.57 | 12.80 | 15.92 | 16.31 |
| Metabolic Energy MJ/kg | 15.00 | 15.04 | 14.56 | 14.48 | 14.06 |
| P/E g/MJ | 31.97 | 32.11 | 36.26 | 36.53 | 41.54 |
| Feed cost yuan/kg | 13.50 | 17.00 | 15.5 | 18.00 | 29.00 |

Notes: LNLP indicates low P/E and low phosphorus feed; LNMP indicates low P/E and middle phosphorus feed; MNLP indicates middle P/E and low phosphorus feed; MNMP indicates middle P/E and middle phosphorus feed; HNP indicates high P/E and high phosphorus feed. The "√" mark in the table indicates the component is present.

### 2.4. Experimental Design and Sample Analysis

The experiment lasted for 8 weeks, from 12 November 2021 to 7 January 2022. Feeding was stopped within 24 h before and after the experiment, and 50 fish were randomly selected from each group and anesthetized with eugenol and weighed to calculate growth efficiency and economic benefits [36]. The water quality experiment was divided into two stages. To determine the change in water quality with time after feeding, the first stage of the experiment was designed to measure the changes in relative concentrations of total ammonia nitrogen (TAN), nitrite nitrogen ($NO_2$-N), total nitrogen (TN), labile phosphate ($PO_4$-P), and total phosphorus (TP) in the water within 6 h after feeding. The second-stage experiment was designed to detect and compare the long-term water quality of each experimental group.

The first stage of the water quality experiment was carried out on the first day of the experiment after weighing the fish. Water samples were collected at the drainage pipe of each fish tank and recorded as the control at 0 h. The fish were then fed after the sampling. Subsequent samples were taken at 0.25 h, 0.5 h, 1 h, 2 h, 4 h, and 6 h after feeding. The concentrations of TAN, $NO_2$-N, TN, $PO_4$-P, and TP in the water were measured, and the concentrations at 0 h were subtracted from these concentrations. The resulting values represent the relative concentration changes of TAN, $NO_2$-N, TN, $PO_4$-P, and TP within 6 h after feeding [37].

The second stage was from day 7 to day 56. The first seven days were spent domesticating the hybrid groupers. Every three days, water samples were collected from the drainage outlet of each fish tank 2 h after feeding for water quality assessment. The measured variables included the TAN, $NO_2$-N, TN, $PO_4$-P, and TP concentrations in the water [38,39].

The concentration of TAN was measured using the hypobromite oxidation method, the concentration of $NO_2$-N was measured by naphthalene ethylenediamine spectrophotometry, the concentration of $PO_4$-P was measured by phosphomolybdenum blue extraction spectrophotometry, and the concentrations of TN and TP were measured using the potassium persulfate oxidation method. The specific experimental operation was performed according to Chinese national standards (GB/T12763.4-2007, GB17378.4-2007).

### 2.5. Calculations and Statistics

#### 2.5.1. Calculation of Growth Efficiency

Growth and feed utilization parameters were calculated as follows [35]:

$$\text{Survival (\%)} = 100\% \times \text{final survival number/initial survival number.}$$

$$\text{Grazing rate (\%)} = 100\% \times (\text{total feeding amount} - \text{residual feed amount})$$
$$/\text{total feeding amount} = 100\% \times \text{total feed intake/total feeding amount}$$

$$\text{Feed coefficient ratio (FCR)} = \text{total feed intake/(final weight} - \text{initial weight).}$$

$$\text{Weight gain rate (WGR, \%)} = 100\% \times (\text{final weight} - \text{initial weight)/initial weight.}$$

$$\text{Specific growth rate (SGR, \% d}^{-1}) = 100\% \times [\ln (\text{final weight}) - \ln (\text{initial weight})]/\text{time.}$$

The economic evaluation of the experimental diet was carried out by calculating the feed cost (unit = yuan) required to produce 1 kg of live-weight fish [40]. The calculation formula is as follows:

$$\text{Feed cost kg}^{-1} \text{ weight gain (FCWG, yuan·kg}^{-1}) = \text{FCR} \times \text{cost kg}^{-1} \text{ feed.}$$

#### 2.5.2. Statistical Methods

SPSS software (version 26) was used to conduct one-way analysis of variance (ANOVA) for growth data and univariate ANOVA (main effects model) for water quality data. Signif-

icant differences in the groups ($p < 0.05$) were determined using Tukey's multiple range test. Values are presented as the mean $\pm$ STDEV (standard deviation).

## 3. Results

### 3.1. Growth Efficiency

The survival rates of all experimental groups were between 98.15 and 99.09%, and the grazing rates were between 96.71 and 98.22% (Table 2). The dietary P/E ratio and phosphorus level had no remarkable effect on survival rate or grazing rate ($p > 0.05$). The weight gain rate and specific growth rate of HNP (97.28 $\pm$ 1.44%, 1.21 $\pm$ 0.01%/d) were highest, and those of MNLP and MNMP were higher than LNLP and LNMP ($p < 0.05$). The feed coefficient of LNLP (1.17 $\pm$ 0.03) was higher than LNMP (1.02 $\pm$ 0.04), while that of LNMP was higher than MNLP (0.81 $\pm$ 0.02) and MNMP (0.80 $\pm$ 0.01) ($p < 0.05$). The feed coefficient of HNP was lowest ($p < 0.05$). The lowest feed cost per kilogram of weight gain was observed for MNLP, while HNP had the highest feed cost per kilogram of weight gain ($p < 0.05$).

**Table 2.** Growth performance and feed utilization efficiency of hybrid grouper (*E. fuscoguttatus* ♀ × *E. lanceolatus* ♂) fed experimental diets for 8 weeks.

| Groups | Survival % | GR % | IW g | FW g | FCR | WGR % | SGR % d$^{-1}$ | FCWG yuan kg$^{-1}$ |
|---|---|---|---|---|---|---|---|---|
| LNLP | 98.15 $\pm$ 1.32 | 98.22 $\pm$ 0.81 | 105.13 $\pm$ 20.04 | 165.87 $\pm$ 36.26 [d] | 1.17 $\pm$ 0.03 [a] | 57.78 $\pm$ 1.34 [c] | 0.81 $\pm$ 0.02 [c] | 16.07 $\pm$ 0.35 [c] |
| LNMP | 98.83 $\pm$ 0.71 | 97.23 $\pm$ 0.92 | 112.40 $\pm$ 20.16 | 180.80 $\pm$ 45.01 [c] | 1.02 $\pm$ 0.04 [b] | 60.93 $\pm$ 3.98 [c] | 0.85 $\pm$ 0.04 [c] | 17.91 $\pm$ 0.70 [b] |
| MNLP | 98.81 $\pm$ 0.60 | 96.71 $\pm$ 1.08 | 110.70 $\pm$ 18.88 | 194.03 $\pm$ 30.50 [b] | 0.81 $\pm$ 0.02 [c] | 77.00 $\pm$ 4.24 [b] | 1.02 $\pm$ 0.04 [b] | 13.00 $\pm$ 0.32 [e] |
| MNMP | 98.31 $\pm$ 0.84 | 97.85 $\pm$ 0.76 | 109.30 $\pm$ 18.03 | 197.23 $\pm$ 36.38 [b] | 0.80 $\pm$ 0.01 [c] | 80.46 $\pm$ 1.59 [b] | 1.05 $\pm$ 0.02 [b] | 14.65 $\pm$ 0.24 [d] |
| HNP | 99.09 $\pm$ 0.92 | 97.58 $\pm$ 0.39 | 105.20 $\pm$ 15.08 | 207.53 $\pm$ 20.38 [a] | 0.70 $\pm$ 0.02 [d] | 97.28 $\pm$ 1.44 [a] | 1.21 $\pm$ 0.01 [a] | 20.92 $\pm$ 0.60 [a] |

Notes: Values are the mean of three replicates $\pm$ STDEV (standard deviation). The different lowercase letters in a row represent significant differences at $p < 0.05$. Means with the same letters or an absence of letters indicate no significant difference between treatments. GR, grazing rate; IW, initial weight; FW, final weight; FCR, feed coefficient ratio; WGR, weight gain rate; SGR, specific growth rate; FCWG, feed cost kg$^{-1}$ weight gain.

### 3.2. Changes in Water Nutrient Pollutants over Time within 6 h after Feeding

The trend in the relative concentrations of nutrient pollutants was roughly similar for the five experimental groups (Figure 2). The relative concentrations of TAN, TN, PO$_4$-P, and TP reached a maximum at 2 h. NO$_2$-N reached a maximum at 1 h and remained at a high level at 2 h. The relative maximum concentrations were TAN, 0.20 mg/L; TN, 4.87 mg/L; and TP, 0.65 mg/L. However, the relative concentrations of NO$_2$-N and PO$_4$-P changed little. The maximum relative concentration of NO$_2$-N was 12.76 µg/L and that of PO$_4$-P was 0.35 µg/L. After reaching their maximums, the relative concentrations of TAN and NO$_2$-N increased again at 6 h, the PO$_4$-P remained at a moderate level, TN and TP continued to decrease to a low level.

### 3.3. Concentrations of Nutrient Pollutants in Water during the Entire Experimental Period

Regarding nitrogen pollutants, the difference between TAN and NO$_2$-N was significantly observed between diets with different P/E ratio levels ($p < 0.05$), the concentrations of TAN and NO$_2$-N were significantly increased by increasing the dietary P/E ratio (Figure 3). It is worth noting that the TAN concentration of the HNP group was much higher than those of the other four groups (21.67~31.64%), and the TAN concentration of the HNP group was close to or exceeded the limit concentration (0.5 mg/L) for part of the time. There was no difference among the five treatments for the total nitrogen concentration ($p > 0.05$). For phosphorus pollutants, there was no significant difference in the concentration of labile phosphate of the water among the five groups ($p > 0.05$). However, significant differences in total phosphorus concentration were observed among the treatments with different dietary available phosphorus levels ($p < 0.05$). The total phosphorus concentration in water of LNLP and MNLP was significantly lower than that of LNMP and MNMP ($p < 0.05$). The total phosphorus concentration of HNP was the highest among the five treatments ($p < 0.05$).

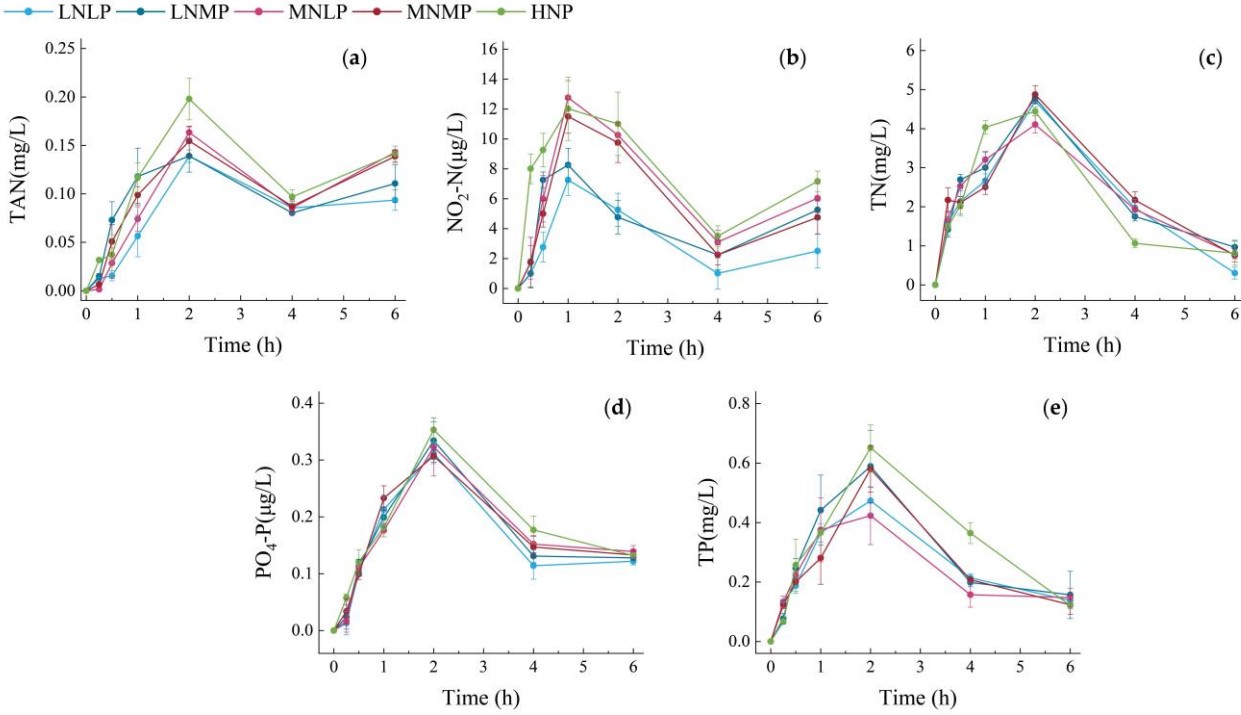

**Figure 2.** Time variations in water quality parameters within 6 h after feeding. (**a**) Total ammonia nitrogen, (**b**) nitrite nitrogen, (**c**) total nitrogen, (**d**) labile phosphate, and (**e**) total phosphorus.

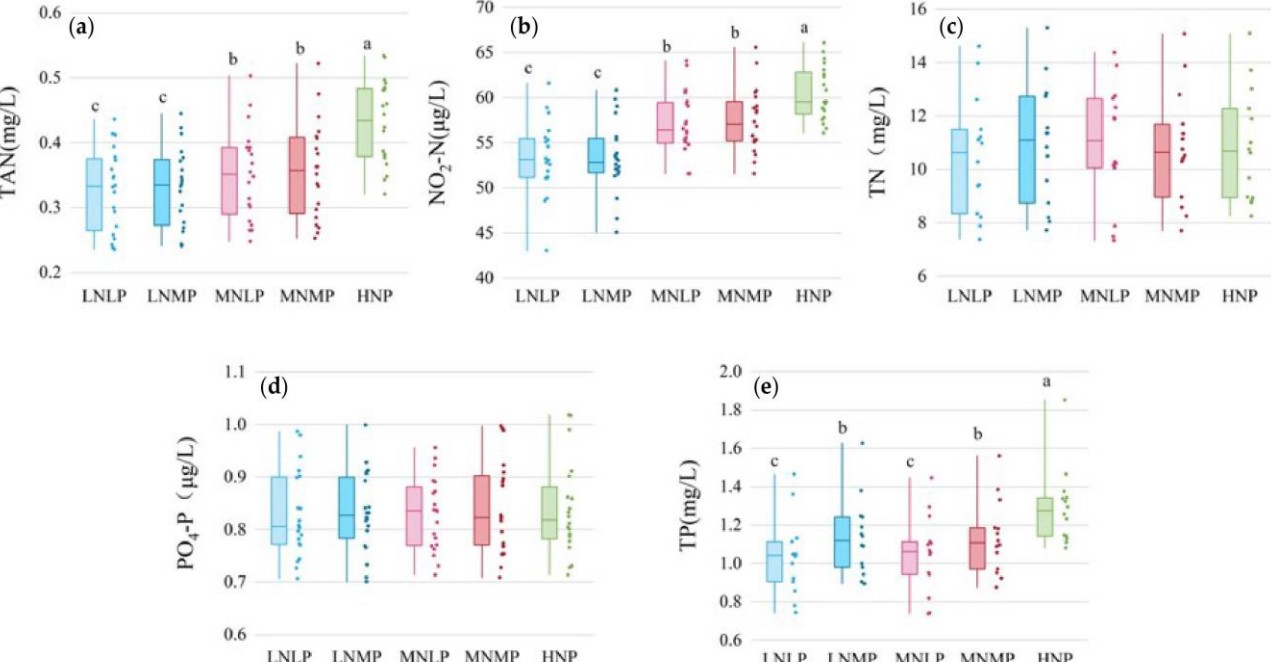

**Figure 3.** Boxplots for the water quality parameters of the five experimental groups within 8 weeks. (**a**) Total ammonia nitrogen, (**b**) nitrite nitrogen, (**c**) total nitrogen, (**d**) labile phosphate, and (**e**) total phosphorus. To eliminate the interference caused by periodic fluctuation of the purification efficiency of the biological filter, the experimental group and time were used as fixed factors, and the effect of feed on water quality parameters was studied using univariate ANOVA (main effects model) and a Tukey's test. The different lowercase letters in the same picture represent significant differences at $p < 0.05$. The absence of letters indicates no significant difference between treatments.

## 4. Discussion

### 4.1. Growth Performance and Feed Utilization

After an 8-week feeding trial, increasing the P/E ratio in the same phosphorus level diet can bring significant growth efficiency advantages (LNLP and MNLP, LNMP and MNMP). However, weight gain rate and specific growth rate were basically at the same level among the diets with the same P/E ratio and different phosphorus levels, and only LNLP and LNMP had significant differences in feed coefficient ratio. It is well known that the increase in dietary P/E ratio can improve the growth efficiency of fish. Jiang et al. (2016) found that feed with the same fat content and high protein level could significantly improve the growth efficiency of hybrid grouper [41]. The study of Gomez-Montes et al. (2003) showed that with the same energy and different protein contents, the growth efficiency of juvenile abalone in the high P/E ratio group was significantly higher than that of the low P/E ratio group [42]. The improvement of growth efficiency with high P/E ratio feed was also proven to apply to Chinese perch, striped perch, rainbow trout, and many other species [43–45]. This is consistent with the results of this study. The increase in dietary phosphorus levels can also increase the growth performance of fish. However, after meeting the maximum phosphorus demand of fish, the increase in dietary phosphorus level could not continue to improve the growth efficiency, such as juvenile *Pelteobagrus fulvidraco* (0.90%), *Lateolabrax maculatus* (1.51%), and *Bidyanus bidyanus* (0.71%) [46–48]. This explained why there was no significant difference in growth efficiency among the experimental groups with the same dietary P/E ratio and different phosphorus levels. The maximum phosphorus requirement of hybrid grouper may be less than or equal to 0.96%. Although the high nutrient feed HNP had the highest WGR, SGR and the lowest FCR, the increase in feed cost resulted in the highest feed cost kg$^{-1}$ weight gain ($20.92 \pm 0.60$ yuan/kg). From the perspective of economic benefits, MNLP ($13.00 \pm 0.32$ yuan/kg) was a better choice in RAS.

### 4.2. The Change of Water Quality after Feeding

In this study, the relative concentrations of TAN and $NO_2$-N both had two peaks, the two peaks of TAN were observed at 2 h and 6 h after feeding, while the peaks of $NO_2$-N appeared at 1 h and 6 h after feeding. These results indicated that in a 20°C RAS, the excretion of dissolved nitrogen waste of hybrid grouper mainly began 0.5 h after feeding and reached its peak at 2 h. Parallel observations were informed in juvenile big-bellied seahorse, *Hippocampus abdominalis* [49]. The peak TAN excretion of big-bellied occurred at 4–6 h and 12–14 h after feeding. In comparison, the relative concentration of $NO_2$-N reached its first peak in 0.5–1 h, which was faster than TAN, and then, remained at a high level at 1–2 h. Nitrite is an intermediate product of the nitrification reaction, in which ammonia is first converted to nitrite and nitrite is then converted to nitrate [50,51]. After feeding, the amount of ammonia nitrogen secreted by fish gradually increased, and nitrite as a byproduct of nitrification reaction also increased [9,52]. When the rate of the nitration reaction reaches its limit, the concentration of nitrite reached its maximum [5,9,53], whereas the fish are still accelerating the secretion of ammonia, and thus, the concentration of TAN continued to rise. This process may explain why the concentration of $NO_2$-N in water reached its maximum faster than that of TAN.

According to the data in this study, the relative concentration variation trend of TN and TP after feeding was basically the same: reached a maximum value at 2 h and then, gradually decreased at a relatively uniform rate. Similarly, $PO_4$-P began to decrease after reaching the maximum value at 2 h, but the relative concentration at 4 h and 6 h was basically at the same level. The RAS used in this study did not have the ability to remove nitrogen and phosphorus, like most of the RAS used in production [54]. TN and TP concentration control was mainly dependent on the exchange with the outside [8,54]. The daily amount of nitrogen and phosphorus waste entering RAS reached equilibrium with the amount of nitrogen and phosphorus waste contained in the water removed by water exchange (10% of the total water in the system). This meant that the concentrations of

TN (7.386–15.297 mg/L) and TP (0.743–1.852 mg/L) in the system would accumulate at higher levels [9,55]. In this study, supplementary water continued to flow into the fish tank at the same rate (12 mL/s). Two hours after feeding, $PO_4$-P, TN, and TP continued to decrease in response to water exchange. The fish continued to import $PO_4$-P into the water through urine [26], and so, the active phosphate concentration did not decrease further after four hours. However, due to the high concentration of TN and TP, although the fish continue to excrete dissolved nitrogen and phosphorus pollutants into the water [26], their concentrations still cannot be prevented from being diluted by water exchange.

*4.3. Effects of Diets with Different P/E Ratios and Phosphorus Levels on RAS Water Quality*

The feed strategy of adjusting the dietary P/E ratio aimed to add nonprotein energy, reduce the consumption of amino acids, and improve the utilization rate of nitrogen. For most fish, reducing the dietary P/E ratio to 18–20 g/MJ can effectively reduce the excretion of dissolved nitrogen waste [26,42]. Based on the water quality results throughout the experimental period, the TAN concentration increased with increasing dietary P/E ratio ($p < 0.05$). A reduction in nitrogen excretion by high levels of dietary P/E ratio was also widely observed in other fish species such as spiny lobster [56], *Litopenaeus stylirostris* [57], and European sea bass [58]. In particular, the TAN concentration of HNP was significantly higher than that in other groups (21.67~31.64%). The gaps between the MN and LN groups were relatively small (6.45–9.10%). This result may be caused by the balance between the rate of ammonia secretion by fish and the rate of ammonia removal by the RAS [9]. After feeding, the fish continuously secreted ammonia nitrogen into the water, which rapidly increased the ammonia nitrogen concentration [59]. At the same time, part of the water in the tank entered the purification system, which reduced the ammonia nitrogen concentration through the biofilm reactors, and then, returned to the tank [9]. When the ammonia nitrogen discharge rate was lower than or close to the system treatment capacity, the two were in dynamic balance, and the ammonia nitrogen concentration was maintained at a stable level [60]. Therefore, there was relatively little difference in the TAN concentration between LN and MN. The reason why TAN concentration in water of HNP group was much higher than that of LN group and MN group may be that the ammonia excretion rate of fish exceeded the nitrogen-loading rates of the system.

Given that the increase in the dietary P/E ratio can significantly promote growth efficiency, the use of feed with a high P/E ratio within the RAS treatment range can improve growth efficiency and ensure that the TAN concentration is maintained at an acceptable level [26]. When the feed P/E ratio causes the excretion of ammonia nitrogen to exceed the treatment capacity of the RAS, it will cause the rapid accumulation of TAN. In this experiment, the average concentration of TAN in the HNP group was 0.431 mg/L, and the TAN concentration exceeded 0.52 mg/L twice and 0.48 mg/L six times. However, China's national standards stipulate that the total ammonia nitrogen in culture water should not exceed 0.52 mg/L at the temperature (20 °C) and pH (8) at which this experiment was conducted [61]. This is not conducive to long-term aquaculture and restricts further improvement of the breeding density of hybrid grouper in RAS.

In this study, there was no significant difference in the concentration of labile phosphate among the five groups of feeds with three different phosphorus levels ($p > 0.05$). It may be that the hybrid grouper requires more phosphorus for growth, and the daily excretion of labile phosphate through urine is low. Previous research showed that the labile phosphate in water mainly comes from dissolved phosphate excreted by fish through urine [9]. When digestible phosphorus levels are below the maximum growth requirement of the fish, only trace amounts of phosphate are eliminated through urine. The maximum growth requirement of phosphorus varies from fish to fish [26]. For hybrid grouper, the regulation of feed phosphorus limited influence on the accumulation rate of labile phosphate under the RAS aquaculture mode, as the growth phosphorus requirement of hybrid grouper is high, and the daily excretion of active phosphate is very small compared with the concentration of active phosphate accumulated in the RAS.

In contrast to that of labile phosphate, the total phosphorus concentration was significantly different between groups with different levels of dietary phosphorus. The TP concentration in the aquaculture water increased with increasing dietary phosphorus levels ($p < 0.05$). This result is partially equivalent to that previously reported by [62], which observed that *Takifugu rubripes*, which has a high demand for phosphorus for growth, the increase in feed phosphorus levels would also increase the concentrations of total phosphorus in aquaculture water. It was speculated that the difference mainly came from the undigested phosphorus in the feed. In addition to dissolved labile phosphate, the total phosphorus in aquaculture water also includes undigested granular phosphorus mainly from feces and feed [63,64]. The data of this study proved that low phosphorus feed could effectively alleviate the accumulation of TP in RAS.

At present, there is little information about the effects of P/E ratio and phosphorus level of feed on RAS water purification capacity, and studies are apparently needed on this research topic.

### 5. Conclusions

In the RAS, the dietary P/E ratio and phosphorus level had significant effects on fish growth and concentrations of TAN, $NO_2$-N, and TP in water. The main nutrient pollutants in water reached the maximum 2 h after feeding, and the nitrite concentration reached the maximum 1 h after feeding. Given the effects of dietary P/E ratio and phosphorus level on growth efficiency, economic benefit, and water quality, feed with an intermediate P/E ratio and low phosphorus level (such as MNLP) is more suitable for long-term use in RAS.

This study provided data support for the management of water nutrient pollutants in RAS and the application of specific feed for RAS.

**Author Contributions:** Conceptualization, X.F. and H.Y.; experiment, X.F.; funding acquisition, H.Y., Z.C. and Z.X.; data curation, X.F.; formal analysis, H.H.; investigation, H.C. and Y.B.; writing—original draft, X.F.; writing—review and editing, X.F., H.Y., Z.C. and H.H.; supervision, Z.C., K.Q. and H.H. All authors have read and agreed to the published version of the manuscript.

**Funding:** This study was supported by the National Key Research and Development Program of China (2020YFD0900603, 2019YFD0900500), Public Welfare Research Plan of Zhejiang Province "Study on the key technology of artemia cultivation using desalination brine" (LGF18D060001), and the Basic Scientific Research Business Fee Project of China Academy of Fishery Sciences (2020TD49). We are grateful to the Yellow Sea Fisheries Research Institute of the Chinese Academy of Fishery Sciences and Huanghai Aquaculture Co., Ltd. for their support with this study.

**Data Availability Statement:** Not applicable.

**Conflicts of Interest:** The authors declare no conflict of interest.

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
