# Peer review of "Optimal Dietary Protein/Energy Ratio and Phosphorus Level on Water Quality and Output for a Hybrid Grouper (Epinephelus lanceolatus ♂ × Epinephelus fuscoguttatus ♀) Recirculating Aquaculture System"

_water, doi:10.3390/w15071261_

Round 1

Reviewer 1 Report

Plagiarism report:

-I have detected 21% similarity through Turnitin software (Please check the attached files). 

Title:

-The title starts with "Effect of" which is reduced the value of the investigation. The author needs to rethink to formulate a new title.

-The author's name and year of discovery of species' scientific names in the title must appear.

Abstract:

-A sentence regarding the implication of the current study can be written at the end of the abstract section.

Keywords:

Please replace the first five keywords, as they already appeared in the title. 

Introduction:

--I found the research gap is written, however, the research questions and a clear hypothesis on this study will be helpful to understand the investigation in more better way. 

-A clear research objective is missing. 

Some highly regarded citations are missing, please add them,

-https://doi.org/10.3390/biology11091288 

-https://doi.org/10.3390/ani12223172

-https://doi.org/10.3390/ani13050887

-https://doi.org/10.3390/ani12091211

Methods:

The methods must be written in the past tense. Most of the sections in the methods did not use any previous citations. As the optimum growth can be observed at 25-30°C for the hybrid grouper, however, this whole study was conducted at 19.2-21.8°C which is the major flow of this investigation. Only for this reason the manuscript can be rejected, but I am considering this can be rechecked. 

Results:

-The result needs to elaborate in the write-up. Especially where figures appeared. As the digit is not shown here, a detailed description of your result will be helpful for the reader to get your findings.

Discussion:

-The discussion section is too boring to read. The reader will lose their concentration. Need to rewrite it in an attractive way, thus the reader will adhere to read until the last sentence. 

-This is not the appropriate way to discuss your result. More papers need to be read. First, you describe your result, then what was the fact of your result, and reason, and then compare it with the previous study. This is the golden rule for writing a discussion. Follow this for other sub-section of your discussion.  

-All the subheadings are not aligned with the presented result section. Either the result sub-heading needs to revise or the discussion sub-heading needs reformation. 

-Many facts and reasons are discussed without any citations in the discussion section! Very strange. 

Conclusions:

-Conclusion must be short, not more than 4-5 sentences. Your experiment is an open-and-shut case, it's very clear. No digit inside. Only very specific findings in the text will appear here. 

-Add: What will be the implication of your study? How farmers will be benefitted. How the government will take your idea in the industry? 

For specific and detailed comments, please check the commented pdf file attached with the upload and below.

L59: Please follow ascending or descending year manner or please follow the journal guidelines. 

L60: However, some countries, for instance, India, Bangladesh, and Indonesia are doing quite well in this technology. Please refer to this here. 

L74: Is it your hypothesis? If not please cite some previous articles. 

L85-88: Please cite,

-https://doi.org/10.3390/fishes7030138

-https://doi.org/10.3354/aei00424

-10.5004/dwt.2011.2761

-https://doi.org/10.1046/j.1355-557x.2001.00027.x

-https://doi.org/10.1038/s41598-019-57063-w

-https://www.researchgate.net/publication/284425485_Nitrogen_and_phosphorus_waste_in_fish_farming

L100: Its a sentence fragment, please complete the sentence. 

L114: The methods must be written in the past tense. Please check the tense in all over the section. 

L117-120: Merge these two sentences. 

L120-121: The sentence is not clear. Please elaborate. 

L126-127: What was the water-holding capacity of these tanks?

L126: What do you mean by this? Please use some digits, for example, you used in the previous sentence. 

L129: Why heating system used? Provide mechanical details. What temperature did you retain in the whole experiment? Mention here. 

L137: This section needs to rewrite. 

L139-140: Please rephrase the sentence. "Cultivated" is not a proper word here. 

L143: Why did you do this?

L140: Why did you do this?

Table 1: What are the elaborations of these abbreviations? Please mention this in the footnote of this table. 

L155-159: Please provide this information in the table. 

L160-163: Where is this? 

L165: NO citation observed. 

L168: You entered the "freshwater" in the system but kept salinity 29.3-31.4 ppt! How? - I think the keyword need to choose more carefully. 

L173-174: This temperature is far below than the optimum growth temperature.

L176: Refer to table number. 

L176: Your experiment is 8 weeks. Why it is written 3 months?

L181: You did not remove the dead fish? Please write it. 

L184: NO citation observed. Please rewrite the whole sub-section. Too many details. Please cite some previous research on the operational procedure. No need for excessive text and all descriptions of your operation. 

L217: NO citation observed. 

L238: The result needs to elaborate in the write-up. 

L252: (Table 2). NO need to write a whole sentence if you can write it simple. Please follow this throughout the MS. 

L252: In methodology, you wrote five treatments considered for this experiment. Now I can see 11 setups. Correct it here or correct it in the methodology. 

L254: is it so? Please check. In the statical analysis section you wrote, "mean of standard error ", please check over the whole MS. 

L271-272: Now it's five treatments. 

L292: Where? Bar? Column? Please be precise. 

L296: -The discussion section is too boring to read. The reader will lose their concentration. Need to rewrite it in an attractive way, thus the reader will adhere to read until the last sentence. 

-This is not the appropriate way to discuss your result. More papers need to be read. First, you describe your result, then what was the fact of your result, and reason, and then compare it with the previous study. This is the golden rule for writing a discussion. Follow this for other sub-section of your discussion.  

L298: The subheading is not aligned to your presented result. 

This is not the appropriate way to discuss your result. More papers need to be read. First, you describe your result, then what was the fact of your result, and reason, and then compare it with the previous study. This is the golden rule for writing a discussion. Follow this for other sub-section of your discussion.  

L299: Its not a very good starting. Please read some other papers, how they start their discussion section. 

L327: The subheading is not aligned to your presented result. 

L339-352: Many facts and reasons are discussed without any citations! Very strange. 

L354-376: Some facts and reasons are discussed here without mentioning any citations. 

L383-392: Some facts and reasons are discussed here without mentioning any citations. 

L395: The subheading is not aligned to your presented result. Follow the previous comment.

L446: Conclusion must be short, not more than 4-5 sentences. Your experiment is an open-and-shut case, its very clear. No digit inside. Only very specific findings in the text will appear here. Add: What will be the implication of your study? How farmers will be benefitted. How the government will take your idea in the industry? 

Author Response

Plagiarism report:

-I have revised the manuscript.

Title:

-I changed the title to: Optimal  dietary protein/energy ratio and phosphorus level on water quality and output for a hybrid grouper (Epinephelus lanceolatus ♂ × Epinephelus fuscoguttatus ♀)  recirculating aquaculture system

-I'm not sure it's appropriate for hybrids to have author and year in the title. I added the author and year in the introduction (L74). If you think this treatment is not suitable, we can continue to modify.

Abstract:

-I have added the meaning of the current research at the end:

Keywords:

-I have re-selected my keywords.

Introduction:

-I have revised and supplemented the introduction and added citations.

Methods:

-I have modified the tenses and added the literature citations.

- I have reviewed the literature and found that in the study by Zhang et al. (2018), hybrid grouper can grow normally when it is above 19℃ (I have added it in the introduction). This experiment was carried out in a temperature-stable factory. Although this experiment is not at the optimum growth temperature of grouper under natural conditions, according to breeding experience in industrial RAS, temperature will not become a limiting factor for hybrid grouper growth when the temperature is above 19℃.

Results:

- I have added data to the text description.

Discussion:

-Thanks for your patient guidance, I have rewritten it.

-I have revised the discussion sub-heading.

- I have added the citation and compared our research with others'.

Conclusions:

- Thanks for your patient guidance, I have condensed the discussion and added the meaning of research.

Please check the attachment for the specific modification

Reviewer 2 Report

Dear Author,

I have read the manuscript "Effects of dietary protein/energy ratio and phosphorus level on water quality and growth efficiency of hybrid grouper (Epinephelus lanceolatus ♂ × Epinephelus fuscoguttatus ♀) in a recirculating aquaculture system" submitted to Water. The concept of the manuscript fits within the aim and scope of the Water Journal. This research focuses on the effect of P/E ratio and phosphorus level on grouper in the RAS system. The results are helpful for the aquaculture industry, especially in indoor culture systems. However, I still have some doubts and comments before publishing. I hope the authors can thoroughly consider and answer my questions below.

Title

Line 7: Please remove the extra asterisk present after the author "Zhengguo Cui".

Introduction

Line 45: The newest aquaculture report of FAO was published in 2022 with the title "The State of World Fisheries and Aquaculture", please update it.

Line 64: The abbreviation of recirculating aquaculture systems is RAS or RASs, please standardize it. Please check the manuscript completely.

Line 66-67: "The system has a considerable impact on water quality." This sentence seems out of place here. Please modify or delete it.

Line 73: The term "sewage" generally refers to domestic wastewater. Here, "aquaculture wastewater" is more appropriate.

Line 85: The sentence "Active phosphate is typically considered a limiting nutrient in the water." What does "Active phosphate" mean? Please provide a definition.

Line 88: It is the first time the term "P/E ratio" is used in the manuscript. Please provide the whole name.

Material and methods

Line 123-125: The author provides the removal efficiency of ammonia and nitrite. Did the author obtain this data from their research or another source? Please provide more information.

Line 127: There is a formatting mistake.

Line 128: Please provide the oxygenation specifications.

Line 154: In Table 1, please provide the standard error and statistical results in the data.

Line 168: Change "Seawater" to "Freshwater".

Line 233: Change "analyses" to "analysis".

Line 234: (P<0.05), P should be in italics.

Results

Line 242: "Three dietary P/E ratios" should be defined in the Materials and Methods.

Line 252: The description of Table 2 is not enough. Please increase it. How does the author define LN, MN, HN, LP, MP, and HP?

Line 271: The pictures in Figure 2 are unclear. Please improve their clarity.

Line 288: If possible, add the limited tolerance line for each nutrient pollutant in the figures.

Line 288: Where is the result of the two-way ANOVA? Please provide it in the form of a table.

Lines 279, 280, 293: The P-value should be in italics. The author states that the significant level is P≤0.05, but this is different from earlier (P<0.05). Please check.

Line 311: What reference does the author use to make comparisons? Please cite.

Lines 431-434: I do not understand what the author wants to explain in this paragraph. Why do water exchange and labile phosphate levels show a positive correlation?

References

The scientific name should be in italics. Please check the references completely.

Author Response

Please see the attachment for specific modification.

Title

- I have removed the extra asterisk.

Introduction

- I have updated the literature citations.

- I have changed the RASs of the whole article to RAS.

- I have rewritten this part.

- Thanks for your suggestion, I have modified it to “aquaculture wastewater”.

- This was a typo. I've changed it to “labile phosphate”.

- I have added the full name of “P/E ratio” here.

Material and methods

- The data is the removal efficiency of ammonia nitrogen and nitrite in the whole experimental period of this study. I have made a supplementary explanation.

- I have modified.

- I have rewritten this part.

- I removed the mean and standard deviation instructions at the bottom of the Table 1.

- Thanks for your suggestion, I have modified it to “seawater”.

- I have made corrections.

- I've set “P” in italics.

Results

- I have rewritten the results and methods.

- I have made a supplement in the methods and reuploaded the Fig.2.

- The results of the two-way ANOVA are shown in Figure 3. I have added to the Notes.

- I have revised the significance level of the whole paper to P < 0.05.

- I have rewritten the discussion and added the citation.

- I have rewritten this part.

References

- I've set the scientific name in italics.

Round 2

Reviewer 1 Report

Dear Editor,

After careful inspection of the given manuscript, some points can be addressed by the authors,

-The hypothesis or research questions can be added to improve the quality

-The implication of the study in the conclusion section and further research approach can be added.

After the above-mentioned corrections have been complied with by the authors, I would like to see a very clean version of the manuscript (The current version is very hard to track and comment). 

Thank you very much.

Author Response

Dear Reviewer,

-The hypothesis have been added to Introduction(L78).

-The implication of the study have been added to Conclusion, and further research approach have been added to Result.

A new version of the manuscript with the Round1 revisions removed has been uploaded

Reviewer 2 Report

Dear Authors,

I have reviewed the revised version of your paper and would like to suggest some improvements to enhance the clarity of the results section and statistical methods section.

Firstly, I would recommend providing a table with the two-way ANOVA results for the two factors and their interaction, including the sum of squares, mean square, F value, and P value. This will help to present the ANOVA results in a clear and concise manner and will also make it easier for readers to understand the relationships between the factors and the observed differences in the data.

Secondly, I suggest that you revise section 2.5.2 Statistical methods to provide more clarity on which parameters belong to one-way ANOVA and which factors are used for two-way ANOVA. This will help readers understand the statistical methods used in the study and the rationale behind the choice of statistical analyses.

I believe that these improvements will enhance the clarity and readability of your paper and help readers better understand your findings.

Author Response

Dear reviewer,

  I made a silly mistake in the previous manuscript.The water quality analysis method we used is not a two-way ANOVA, it's univariate ANOVA (main effects model). We have updated the full article (2.5.2; L212). I set time and experimental group as fixed factors, and water pollutant concentration as dependent variable to get the difference of water quality concentration between the groups without the interference of time factor. Since the interaction of time and groups is not present in univariate ANOVA, we believe that adding a table does not complement Figure 3.

Round 3

Reviewer 2 Report

Dear authors, 

Please check the references section again to ensure that this formatting error is corrected. e.g.

The scientific name should be italics.

The year should be in bold font....

Then, I suggest that rewrite the keywords section to make it more attractive and informative. It would be beneficial to provide 4-6 relevant keywords that accurately reflect the topics covered in your paper.

Author Response

Dear Reviewer,

-I have corrected the format.

-I have re-selected the keywords.
